# Vesiculation of Red Blood Cells in the Blood Bank: A Multi-Omics Approach towards Identification of Causes and Consequences

**DOI:** 10.3390/proteomes8020006

**Published:** 2020-03-31

**Authors:** Joames K. Freitas Leal, Edwin Lasonder, Vikram Sharma, Jürgen Schiller, Giuseppina Fanelli, Sara Rinalducci, Roland Brock, Giel Bosman

**Affiliations:** 1Department of Biochemistry (286), Radboud University Medical Center, 6500 HB Nijmegen, The Netherlands; joames.leal@gmail.com (J.K.F.L.); roland.brock@radboudumc.nl (R.B.); 2Department of Applied Sciences, Faculty of Life and Health Sciences, Northumbria University, Newcastle-Upon-Tyne NE1 8ST, UK; edwin.lasonder@northumbria.ac.uk; 3School of Biomedical Sciences, University of Plymouth, Plymouth PL4 8AA, UK; vikram.sharma@plymouth.ac.uk; 4Institute for Medical Physics and Biophysics, Medical Faculty, University of Leipzig, 4107 Leipzig, Germany; juergen.schiller@medizin.uni-leipzig.de; 5Department of Ecological and Biological Sciences (DEB), University of Tuscia, 01100 Viterbo, Italy; giuseppina.fane@gmail.com (G.F.); sara.r@unitus.it (S.R.)

**Keywords:** red blood cell, vesicles, proteomics, lipidomics, metabolomics, storage, aging

## Abstract

Microvesicle generation is an integral part of the aging process of red blood cells in vivo and in vitro. Extensive vesiculation impairs function and survival of red blood cells after transfusion, and microvesicles contribute to transfusion reactions. The triggers and mechanisms of microvesicle generation are largely unknown. In this study, we combined morphological, immunochemical, proteomic, lipidomic, and metabolomic analyses to obtain an integrated understanding of the mechanisms underlying microvesicle generation during the storage of red blood cell concentrates. Our data indicate that changes in membrane organization, triggered by altered protein conformation, constitute the main mechanism of vesiculation, and precede changes in lipid organization. The resulting selective accumulation of membrane components in microvesicles is accompanied by the recruitment of plasma proteins involved in inflammation and coagulation. Our data may serve as a basis for further dissection of the fundamental mechanisms of red blood cell aging and vesiculation, for identifying the cause-effect relationship between blood bank storage and transfusion complications, and for assessing the role of microvesicles in pathologies affecting red blood cells.

## 1. Introduction

In the blood bank, red blood cells (RBCs) are exposed to various stressful circumstances that affect their structure and function. The accumulated effect of these stressors consists of changes in cell shape [1], reduction in deformability [2,3], an increased tendency to aggregate [4], enhanced adherence to endothelial cells [5,6], and increased susceptibility to phagocytosis and hemolysis [7]. At the molecular level, there is an increase in removal signals on the RBC surface, caused by the generation of senescent cell antigens [8,9], the exposure of phosphatidylserine [10,11], and a decrease in concentration and/or activity of CD47 [12,13]. Associated—and possibly causative—processes are the accumulation of oxidized proteins and lipids, increased protein breakdown, depletion of ATP, alterations in ion concentrations, and disturbed phospholipid organization [14]. All these phenomena are accompanied by the formation of microvesicles [15,16], making microvesicle generation an important consequence of RBC aging in vitro.

Vesiculation has been postulated to protect against untimely removal of RBCs in vivo, by the shedding of cell membrane patches that have been damaged by aging-associated processes [17]. However, RBC-derived microvesicles contain hemoglobin, which, in pathological conditions, may enhance iron accumulation in the heart and the kidneys, and the removal signals on microvesicles may promote coagulation, inflammation, and autoimmune responses [18,19,20].

Even though data on RBC-derived vesicles in the circulation are scarce, they nevertheless show that these vesicles share essential features with those of RBC concentrates, suggesting that the same processes occur during RBC aging in vitro and in vivo [21,22]. Thus, the identification of the mechanism responsible for microvesicle generation in the blood bank may not only be instrumental in the reduction of transfusion side effects, but also in the elucidation of the fundamental mechanisms of RBC aging in vivo. The increasing awareness that alterations in RBC homeostasis affect the whole organism emphasizes the clinical value of molecular knowledge on RBC aging [23,24].

The presently available data for RBC concentrates show that the composition of microvesicles changes with storage time, but the extent and the cause–effect relationship between these changes remain unclear [22,25]. Therefore, we used a multidisciplinary approach combining immunochemical, morphological and multi-omics analysis for a detailed study of the changes that microvesicles undergo during storage in the blood bank. Our results indicate that different mechanisms become prominent at different storage times. Additionally, these results constitute a comprehensive, integrative data set for further elucidation of the causes and consequences of RBC vesiculation.

## 2. Materials and Methods

Microvesicles (MVs) were obtained from five standard RBC concentrates of anonymous donors of the regional blood bank Sanquin Bank South East Region, Nijmegen, The Netherlands, following the guidelines of the local medical ethical committee and in accordance with the declaration of Helsinki. At each storage time (one, three, and five weeks), the MVs were isolated from 10 mL samples as described before [26]. All analyses were performed on at least three biological replicates.

MVs were analyzed by flow cytometry using mixtures of FITC-labeled CD235a (1:100, clone 11E4B-7-6, Beckman Coulter, Fullerton, CA, USA) and Annexin V-PE (1:25, BD Pharmingen, Hoeven, the Netherlands), and with rabbit anti-human band 3 (1:1000, clone K2N6B/PM) combined with secondary antibody goat anti-rabbit Alexa 633 (IgG, A 21070, Rockford, USA). CD59 content was determined using PE-labeled CD59 (clone MEM43, 1:400, IQ products, Groningen, the Netherlands) in combination with FITC-labeled CD235a. Sulfate latex microspheres (0.9 μm; Invitrogen, Carlsbad, CA, USA) and washed Flow-Count calibration beads (Beckman Coulter, Brea, CA, USA) were used for quantification [26].

Cryo-transmission electron microscopy (Cryo-TEM) was performed using a JEM 1400 Flash electron microscope (120KV, Jeol, Tokyo, Japan; Nieuw-Vennep, The Netherlands) as described before [27]. To obtain more detailed structural information, some samples were imaged using bright field phase contrast transmission electron microscopy at 200 kV on a Talos Arctica microscope with a Falcon II direct electron detector and Volta phase plate with a total dose of about 100e-/Å^2^ in linear mode. Volta phase plate imaging was performed in focus. Samples were plunge-frozen into liquid ethane cooled by liquid nitrogen to just above its freezing point, with some solid ethane kept in equilibrium with the liquid. The plunging was performed on a Thermo Fisher Vitrobot Mark IV using 2.5 s blot time, 4 °C, 100% relative humidity, and the blot force adjusted to have the blot pads just touch without blot paper. Quantifoil R1.2/1.3 holey carbon grids were used, and 2.5 μL of the sample was applied just before blotting.

For proteomic analysis, MV samples were processed using a filter-aided sample preparation (FASP) procedure as described previously [28], and enzyme digests were acidified to a final concentration of 0.1% TFA and purified by STAGE tips [29]. LC-MS/MS experiments were carried out by the LC-MS/MS platform composed of an Ultimate 3000 UPLC (Thermo Fisher, Dreieich, Germany) connected to the Orbitrap Velos Pro mass spectrometer (Thermo Fisher, Germany). Peptides were loaded on a 2-cm Acclaim PepMap100 Nano-Trap Column (Thermo Fisher, Germany) and separated by a 25-cm Acclaim PepMap100 Nano LC column (Thermo Fisher, Germany) packed with 3-μm C18 beads. The flow-rate was set at 300 nl/min in a 120 min gradient of 95% buffer A/5% buffer B to 65% buffer A/35% buffer B (buffer A: 0.5% acetic acid; buffer B: 0.5% acetic acid in 100% acetonitrile). Peptides eluting from the column were electrosprayed into the mass spectrometer at 2.3 kV spray voltage. MS data acquisition was set in a data-dependent mode, automatically switching between MS and MS2, where full-scan spectra of intact peptides (*m*/*z* 350–1500) were acquired with an automated gain control accumulation value of 1,000,000 ions. The 10 most abundant ions were sequentially isolated and fragmented in the C trap, where dissociation was induced by HCD mode, using an accumulation target value of 10,000, a normalized collision energy of 45%, and a capillary temperature of 275 °C. Dynamic exclusion of ions sequenced within the 45 previous seconds was applied. Unassigned charge states and singly charged ions were excluded from sequencing. For MS2 selection, a minimum of 10,000 counts was required.

The MaxQuant—Andromeda search engine integrated in the MaxQuant software (Version 1.3.0.5) was used for identifying proteins [30,31]. Peak lists were generated for the top 12 most intense MS peaks in 100 Da windows by MaxQuant prior to the human UNIPROT database (release 2017/04) search. The protein database was supplemented with frequently observed contaminants from MaxQuant. Andromeda search parameters for protein identification were set to a tolerance of 6 ppm for the parental peptide and 0.5 Da for fragmentation spectra and trypsin specificity allowing up to 2 miscleaved sites. Deamination of glutamine, oxidation of methionine, and protein N-terminal acetylation were set as variable modifications, carboxyamidomethylation of cysteines was specified as a fixed modification. Minimal required peptide length was specified at six amino acids. The ‘match between run option’ for an elution time window of two minutes was enabled. Peptides and proteins detected by at least two peptides in one of the samples with a false discovery rate (FDR) of 1% were accepted. Excluded from validation were proteins identified by site only, external contaminants, and reversed proteins. Proteins were quantified by normalized summed peptide intensities [32], computed in MaxQuant with the label-free quantification (LFQ) option switched on. The mass spectrometry proteomics data have been deposited to the ProteomeXchange Consortium (http://proteomecentral.proteomexchange.org) via the PRIDE partner repository with the dataset identifier PXD017056.

Microvesicle lipids were analyzed by matrix-assisted laser desorption and ionization time-of-flight (MALDI-TOF) mass spectrometry as described before [33,34].

The metabolomes of stored RBC microvesicles were charted as described before [35]. Raw files of replicates were exported, converted into mzXML format through MassMatrix (Cleveland, OH, USA), and then processed by MAVEN software (Version 8.0, http://maven.princeton.edu/). Metabolites were graphed with Graphpad Prism 5.01 (Graphpad Software, La Jolla, USA) and one-way analysis of variance (ANOVA, *p* < 0.05) followed by a Bonferroni post-hoc test performed using the same software.

## 3. Results

### 3.1. Morphology by cryo-EM

The formation of microvesicles (MVs) is an integral part of the aging process of the RBC in vitro. To identify aging-related changes in RBC derived vesicles, MVs were isolated from blood bank units after 1, 3, and 5 weeks of storage. These sampling times were selected mostly based on previous observations showing that most irreversible changes in morphology occur around 21 days of storage [1]. We found a considerable increase in the concentration of RBC-derived MVs during storage from 4300 ± 1000 MVs/μL in week one and 7400 ± 5400 MVs/μL in week three to 25,400 ± 17,600 MVs/μL in week five, confirming previous observations, also regarding the large interindividual variation [1,10,15]. Detailed electron microscopic analysis showed that the MVs became smaller with storage time, from a diameter of approximately 160 nm after week one and three to approximately 145 nm after five weeks of storage. Microvesicles are enriched in glycated and oxidized hemoglobins that are concentrated in denatured hemoglobin aggregates that are distinguished by a higher electron density [16,36]. We found that these aggregates accumulate especially during the first three weeks of storage (Figure 1). The fraction of aggregate-containing vesicles increases significantly from 24 ± 16% (*N* = 141) in week one to 55 ± 13% in week three and five (*N* = 239; *p* < 0.05). Interestingly, approximately 15 percent of the microvesicles contain a smaller vesicle (Figure 1B,C). This was confirmed by a more detailed analysis (Figure 1C).

### 3.2. Membrane Organization by Flow Cytometry

Vesiculation is, in general, associated with changes in phospholipid organization, and most MVs expose phosphatidylserine (PS) [17,18,26]. During storage, the fraction of PS-positive MVs increased from approximately 70% in week one to almost 100% in week three and week five (Figure 2A). Notably, the concentration of phosphatidylserine was much higher in the three-week-old MVs than before or after that period, as indicated by the changes in mean fluorescence intensity (Figure 2B). As for the morphological data, we observed a relatively large heterogeneity in PS exposure and band 3 content. These data provide the first indications that the mechanisms underlying microvesicle generation may change with storage time.

Changes in band 3 may play a major role in microvesicle generation [16,17,38,39]. Flow cytometry analysis showed that the fraction of band 3-containing MVs, as well as the band 3 content per microvesicle, decreased with storage time (Figure 2C,D).

Storage is accompanied not only by changes in membrane organization and protein content, but also by changes in protein function, as exemplified by our recent observations on the activity of GPI-linked acetylcholinesterase [40]. Additionally, changes in GPI-linked proteins have been suggested to be involved in the vesicle generation mechanism [37,41]. Therefore, we analyzed the presence of the GPI-linked CD59, the inhibitor of complement-induced lysis. Our data show that almost all MVs contained CD59 after the first week of storage, but that the population of CD59-positive MVs decreased significantly later during storage (Figure 2E).

### 3.3. Phospholipid Composition by Mass Spectrometry

The storage-associated changes in phosphatidylserine exposure on RBCs [11,42] and microvesicles (Figure 1) suggest that changes in the lipid component of the RBC membrane may be part of the mechanism of microvesicle generation. To obtain more direct information on lipid composition, we subjected the lipid fraction of microvesicles to mass spectrometry. Our analyses showed no changes in microvesicle phospholipid composition, except for an increase in lyso-phosphatidylcholine after two weeks of storage (Figure 3).

### 3.4. Storage-Related Changes in the Vesicular Metabolome

Based on various descriptions about the relationship between RBC membrane organization and activity of key glycolytic enzymes [43,44,45], we postulated that the storage-associated changes in RBC membrane organization and associated disturbances in the kinetics of energy metabolism should be reflected in the microvesicle metabolome.

Our data show that the vesicular concentrations of most metabolites of the central RBC pathways, such as the glycolysis (3-phospho-glycerate) and the pentose phosphate pathway (xylulose 5-phosphate), did not change with storage time (Figure 4). However, the late intermediates of these pathways, such as ribose-5-phosphate in the pentose phosphate pathway and phosphoenolpyruvate of glycolysis, were significantly decreased in five-week-old microvesicles. Moreover, we found a strong decrease in lactate in the three- and five-week-old vesicles. Interestingly, the vesicular concentrations of the ATP-related metabolites inosine and hypoxanthine show the kinetics observed for the RBC cytoplasm during storage (Figure 4C), where energy depletion and a storage-induced activation of the purine salvage pathway has been observed [46]. Most of the amino acids in the vesicles increased in level after three weeks of storage. However, except for lysine and arginine, which further increased to week five, their concentration decreased in five-week-old microvesicles (Figure 4D).

All data are represented as the means of three biological and three technical replicates ± SEM. One-way ANOVA (*p* < 0.05), followed by a Bonferroni post hoc test (alpha level 0.05) was used to determine pairwise statistically significant differences. If two variables have different letters, they are significantly different from each other; for all variables with the same letter, the difference between the means is not statistically significant. G6P, glucose 6-phosphate; F1,6BP, fructose 1,6 bisphosphate; GA3P, glyceraldehyde 3-phosphate; 3PG, 3-phosphoglycerate; PEP, phosphoenolpyruvate; 6PG, 6-phosphogluconate; Ru5P, ribulose 5-phosphate; Xu5P, xylulose 5-phosphate; R5P, ribose 5-phosphate; Sed7P, sedoheptulose 7-phosphate; E4P, erythrose 4-phosphate; PRPP, phosphoribosyl pyrophosphate; HPX, hypoxanthine.

### 3.5. Storage-Dependent Changes in the Microvesicle Proteome

Previous proteomic analyses have shown storage-associated changes in the proteomes of RBCs and RBC-derived microvesicles [16,22]. Here, we performed an analysis of the MV proteome to place the proteome into the context of our other analyses. From the 198 identified proteins (see Materials and Methods), we selected some key markers. The cytoplasmic protein hemoglobin, the membrane protein band 3 and the plasma-derived IgG all increase with storage time (Figure 5). Band 3 is highly enriched in microvesicles, in comparison with other integral membrane proteins such as the glucose transporter GLUT2 (Figure 5, Appendix A). It is noteworthy that this enrichment is also observed for the microdomain-associated proteins stomatin and flotillin-1 and flotillin-2 (Appendix A). Overall, we could distinguish two patterns in the kinetics of vesicle enrichment. Vesicular enrichment either showed a maximum at week 3 followed by a decrease towards week five, or a continuous increase over the whole period. Even though the changes varied in extent, the overall patterns showed good reproducibility between the three biological replicates (Figure 5 and Appendix A). The protein abundance patterns of the membrane protein band 3 and the cytoskeleton protein spectrin show conspicuous alterations in microvesicle membrane composition during storage (Figure 5). The band 3/spectrin ratios show an approximate 10-fold enrichment of band 3 in the microvesicles relative to the RBC membrane between week 3 and week 5 [16,22], and the band3/GLUT2 and band 3/actin ratios show an even stronger enrichment of band 3 already during one week of storage from five-fold for the band 3/GLUT2 and up to 100-fold for the band 3/actin ratio; Appendix A). The signals for IgG and complement show similar patterns, with a most pronounced increase after three weeks of storage (Figure 5). In contrast, the band 3/stomatin ratio decreases up to 10-fold with storage time (Appendix A).

## 4. Discussion

Our integrated morphological and metabolic analysis of microvesicles generated during storage demonstrates that microvesicle composition changes on all levels.

### 4.1. Morphology (cryo-EM)

During the storage of RBC concentrates in the blood bank, the number of microvesicles increases strongly, especially between three to five weeks [1,10,15]. Our morphological analyses show high heterogeneity in the size and shape of these microvesicles at all time points, and that their morphology changes with storage time (Figure 1). These data confirm previous findings of a gradual increase in size from approximately 140 to 160 nm obtained in fixed microvesicle preparations, and data suggesting the presence of degenerated membrane patches [47]. A small number of these vesicles may be exosomes originating from the few reticulocytes that were present in the blood at the time of collection, which mature during the first days of storage [14,16]. Many vesicles contain electron-dense material, probably consisting of aggregated hemoglobin [22,47]. The presence of the extracellular hemoglobin-binding and heme-binding proteins haptoglobin and hemopexin in the microvesicle proteomes (Appendix A) confirms electron microscopic indications for the shedding of intravesicular material, especially after prolonged storage [47]. Surprisingly, we observed smaller vesicles inside 15 percent of the microvesicles in all samples (Figure 1). Such intravesicular vesicles were found also in freshly isolated samples, making it unlikely that their presence is an artifact caused by the sample preparation. These observations require more detailed investigation but suggest that during storage, when microvesicles are not removed as in the circulation, vesiculation may proceed within existing vesicles. Together with the biphasic patterns of many of our present observations, these data support the hypothesis that various vesiculation mechanisms become active with increasing storage time, with a critical period around three weeks of storage [1,14,16,19,22,48].

### 4.2. Membrane Organization 1: Flow Cytometry and Phospholipid Analysis

Quantitative analysis of PS exposure did not only show an increase in PS-exposing microvesicles during storage as described before [13,19,42], but also changes in the PS content with the highest concentration in microvesicles in three-week-old concentrates (Figure 2). Although we found no statistically significant correlations between microvesicle size and PS (or any other surface markers), these data indicate that changes in microvesicle morphology are associated with changes in the phospholipid component of the RBC membrane. Such changes likely consist of changes in the organization rather than changes in composition, as suggested by mass spectrometry showing a rather stable phospholipid composition (Figure 3). In agreement with other observations [49], we observed a small increase in SM 16:0 and a small decrease in SM 24:1, which may be associated with increased vesicle formation [34]. Additionally, we find an increase in lyso-PC upon prolonged storage (Figure 3). Lyso-PC and lyso-PE levels are increased after RBC rejuvenation *in vitro*, probably as a consequence of cell damage [50]. Previously, we have observed an increase in the lyso-PC content of the RBC membrane in patients with sepsis, together with increased microvesicle numbers and increased phospholipase A2 activity in the blood [34]. Additionally, we have found that lyso-PC is present in vesicles, but not in RBC membranes [48]. These findings all suggest that small changes in phospholipid composition of the RBC membrane, resulting in changes in organization, may be involved in the generation of microvesicles in affected, susceptible membrane regions. We postulate that the strong increase in apolipoprotein A content with storage (Figure 5), as has also been observed for apolipoprotein J [51], may be the result of the emergence of new phospholipid-binding sites. The concomitant appearance of PS and lyso-PC promotes microvesicle recognition and removal [51,52,53,54].

### 4.3. Membrane Organization 2: Flow Cytometry and Proteomics

Proteomic data on the storage-associated increase in the lipid raft-associated proteins stomatin and flotillin in microvesicles supports the involvement of the lipid-involving membrane reorganization in microvesicle formation [16,41,55]. The kinetics of the content of these proteins in microvesicles show a biphasic pattern, as observed for integral membrane proteins such as band 3 (Figure 5). The binding of stomatin and/or flotillin to microdomains, in combination with oligomerization and binding to band 3 complexes [55,56,57], may affect the organization of the lipid part of the cell membrane and the stability of the spectrin/actin cytoskeleton. This will disturb the balance between the forces that determine membrane curvature, and result in evagination and vesicle formation [58]. Interestingly, the band 3/stomatin ratio decreased 10-fold after the first week of storage, whereas the band 3/spectrin and band 3/actin ratios increase 10-fold and six-fold, respectively. This may be caused by a weakening of the binding of band 3 to the ankyrin/spectrin binding site [22,55], leading to an increase in monomeric band 3 molecules. The increased mobility of band 3 monomers [59] could result in the enrichment of band 3 in the microvesicles. Interestingly, our proteome analysis shows an increase in band 3, whereas flow cytometry analysis shows a storage-dependent decrease in band 3 exposure (Figure 3). These results support the hypothesis that storage-associated changes in band 3 conformation affect the interaction between membrane and cytoskeleton especially at the band 3/ankyrin complex, and that the resulting microvesicles serve to remove damaged proteins from the aging RBCs [17,22]. In addition, changes in band 3 conformation may contribute to the fast removal of microvesicles by an increase in senescent cell antigen-binding IgG. Similar indications for storage-associated changes in membrane protein conformation as well as organization are provided by the enrichment of the GPI-linked proteins CD55, CD59, and acetylcholinesterase on microvesicles (Figure 5). There are indications that the microvesicle forms of these proteins are inactive [37,40]. Accumulation of inactive CD55 and CD59 is the likely cause of the accumulation of complement (Figure 5). Together with the exposed phosphatidylserine (Figure 2) and the microvesicle-bound IgG (Figure 5), this will contribute to the fast removal of the microvesicles from the circulation. However, flow cytometric, as well as proteomic data, show that microvesicles without CD55 and CD59 accumulate at longer storage times (Figure 4 and Figure 5). The concomitant accumulation of complement, in combination with the accumulation of proteins such as prothrombin, antithrombin, plasminogen, and IgG (Figure 5), indicates that removal of microvesicles or prevention of their generation will reduce the unwanted side effects of transfusion. This may be especially relevant when the physiological clearing system is overwhelmed by large microvesicle numbers, e.g., in transfusion-dependent patients or after transfusion of older RBC concentrates. The large interindividual heterogeneity we observed in quantity of protein and PS expression (Figure 1 and Figure 2) suggests that microvesicle analysis may be a useful tool to select the least-harmful red blood cell concentrates for transfusion, especially in vulnerable patients.

### 4.4. Metabolome Changes as Clues to Vesiculation Mechanisms

Red blood cell aging in vivo and in vitro is accompanied not only by changes in morphology, function, and structure, but also by changes in metabolism. Some of these changes, such as decreases in ATP and 2,3-DPG, are part of the blood bank quality control system [60]. Recently, metabolome analyses have revealed many aging-associated changes in RBC metabolism [46,50]. These changes may reflect alterations in enzyme and/or membrane transport activity. Glycolytic enzymes are organized in multi-protein complexes that bind to band 3 and/or various other membrane proteins, and changes in band 3 conformation regulate their activity [44,45,61,62,63]. Metabolomic data of misshapen RBCs support the hypothesis that alterations in membrane protein conformation affect various metabolic pathways [2,64,65,66]. Therefore, we postulated that the metabolome of microvesicles could reveal changes in membrane protein organization. Our present data show that the kinetics of the concentrations of glycolysis intermediates reflect the microvesicle content of glycolysis enzymes (Figure 4 and Figure 5). Additionally, the changes in the microvesicular concentration of most metabolites during storage mirror the changes in the parent RBCs [46]. One notable exception is the strong decrease in lactate in the microvesicles during storage, which is in marked contrast to the increase in RBCs and in the supernatant [46]. This observation emphasizes the involvement of a carrier-dependent pathway in the transport of lactate across the RBC membrane. Interestingly, the microvesicles from five-week-old RBC concentrates contained very high levels of hypoxanthine compared with the one-week-old microvesicles (Figure 4). Recently, the accumulation of hypoxanthine has been identified as a metabolic marker of storage lesions with clinical implications, which upon conversion by xanthine oxidase generates reactive oxygen species [67,68,69,70]. Our findings suggest that, next to old RBCs, microvesicles may be the main vehicle for this toxic molecule.

The amino acids displayed various accumulation kinetics during storage, varying from an increase after three weeks followed by a decrease at five weeks for the aromatic amino acids phenylalanine, tryptophan, and tyrosine, to a strong increase over time for lysine and arginine, whereas the concentrations of valine and cysteine hardly changed during the storage period (Figure 4). These data confirm metabolomic indications for aging-related and cell morphology-associated effects on the activity of various transport systems in RBCs [65,66]. Additionally, these data indicate that in-depth knowledge of the changes that the metabolism and transport of amino acids, as well as the activity of proteases that the RBC undergo during aging and storage, will be relevant for understanding the role of RBCs in maintaining organismal homeostasis [23,24].

## 5. Conclusions

Our data emphasize the close interaction between the lipid and protein compartments of the membrane in maintaining an intact, functional RBC. However, the relatively small number of donors and samples that could be analyzed with all techniques (morphology, removal markers, lipidomics, proteomics, and metabolomics) precludes a conclusive correlation analysis of our microvesicle data with the regular RBC blood bank markers such as pH, ATP, etc. A few hypotheses can be postulated on the cause/effect relationship between the changes in the lipid, protein, and metabolite composition of microvesicles, leading to an integrative mechanism of vesicle formation. In a lipid-centered hypothesis, a small increase in lyso-PC and/or decrease in sphingomyelin is the initial event that could induce a weakening of the band 3/ankyrin/spectrin linkage. The recruitment of stomatin to cholesterol/ceramide-rich domains could enhance the curvature of the membrane at these sites. Loss of interactions with other proteins would then lead to changes in band 3 conformation. In this scenario, removal or inhibition of phospholipase or sphingomyelinase activity from the RBC concentrates should have a pronounced effect on microvesicle numbers and composition [71]. Our data do not support this hypothesis, as the largest increase in the lyso-PC content of the microvesicles, and the increase in the lipid-anchored acetylcholinesterase occurs especially after three weeks of storage. Thus, changes in phospholipid organization, if they play a causal role in microvesicle generation at all, are likely to be secondary processes.

In a protein damage-centered hypothesis, vesiculation starts by changes in band 3 conformation that weaken the band 3/ankyrin/spectrin linkage. The high concentrations of catalase and peroxiredoxin (Figure 5) support the hypothesis that oxidative damage is the main mechanism for the aging-related changes in band 3 conformation, either directly or by an effect on the band 3/ankyrin/spectrin linkage [17,22]. Thus, oxidative damage-inducing changes in band 3 conformation may be the primary trigger for vesiculation. Various observations support this hypothesis. The involvement of oxidative damage is supported by the reduction in microvesicle formation upon the addition of ascorbic acid to the storage medium [72]. The specific increase in vesicular band 3 content during the first three weeks of storage (Figure 5) and the concomitant decrease in band 3 antigenic activity (Figure 2) support the relationship between changes in conformation with a decrease in binding to the cytoskeleton. The changes in vesicle morphology (Figure 1) provide further evidence for the presence of progressive alterations in the interaction between the membrane and the cytoskeleton. Mechanical investigations, showing that vesicles from fresh RBCs are softer than vesicles from stored RBCs and that this is mainly dependent on the protein/lipid ratio [48], support the hypothesis that small changes in protein organization are the main triggers of vesicle generation. Our findings on the content of the microdomain-associated protein stomatin and the GPI-linked proteins (Figure 2, Figure 5) indicate that small, local changes may trigger secondary changes in lipid organization.

## Figures and Tables

**Figure 1 proteomes-08-00006-f001:**
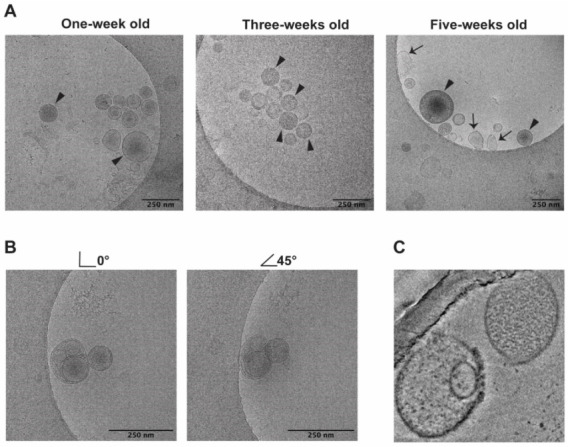
Cryo transmission electron microscopy images of microvesicles formed during RBC storage. (**A**) the arrow heads point to microvesicles with electron-dense material; the arrows point to less rounded, misshapen microvesicles; (**B**), images of a sample at 0° and tilted at 45°, showing vesicles inside other vesicles; (**C**), more detailed images of a sample obtained with a Talos Arctica electron microscope (see Materials and Methods). All images were obtained from the same RBC concentrate.

**Figure 2 proteomes-08-00006-f002:**
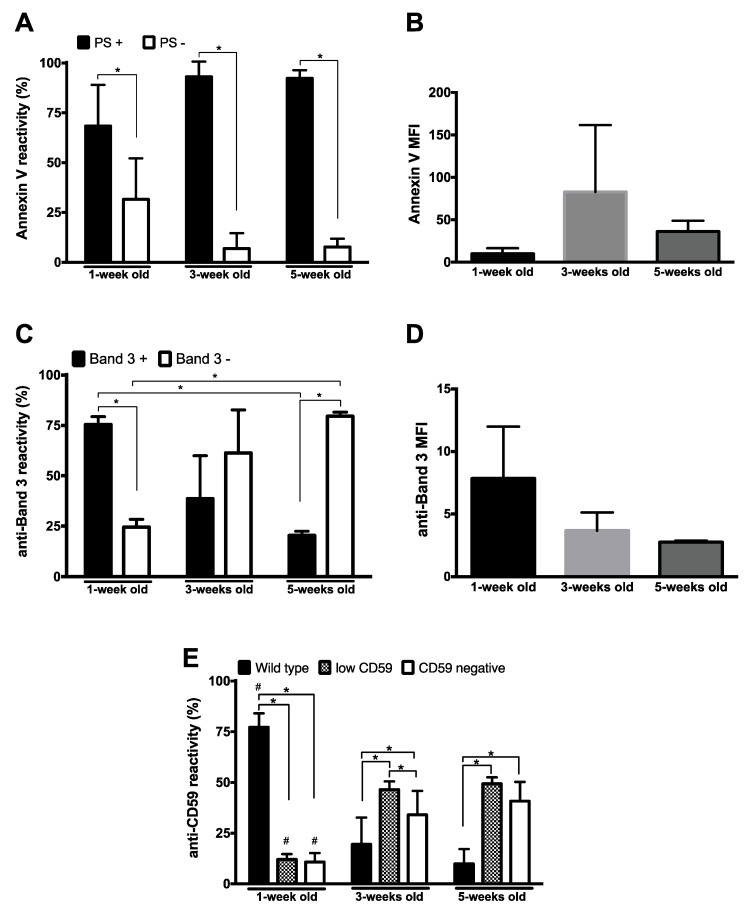
Aging markers on microvesicles during storage. (**A**), percentage of RBC-derived MVs reactive with Annexin V (PS+); (**B**), Annexin V mean fluorescence intensity (MFI). (**C**), percentage of RBC-derived MVs stained by anti-band 3 antibody; (**D**), anti-band 3 mean fluorescence intensity; (**E**), percentage of CD59-reactive MVs. RBC-derived MVs were categorized into wild type, CD59 low, and CD59-negative, as described before [37]. #,* *p* < 0.05 (*N* = 5), based on a comparison of the percentages. RBC-derived vesicles were isolated and analyzed by flow cytometry as described in Materials and Methods.

**Figure 3 proteomes-08-00006-f003:**
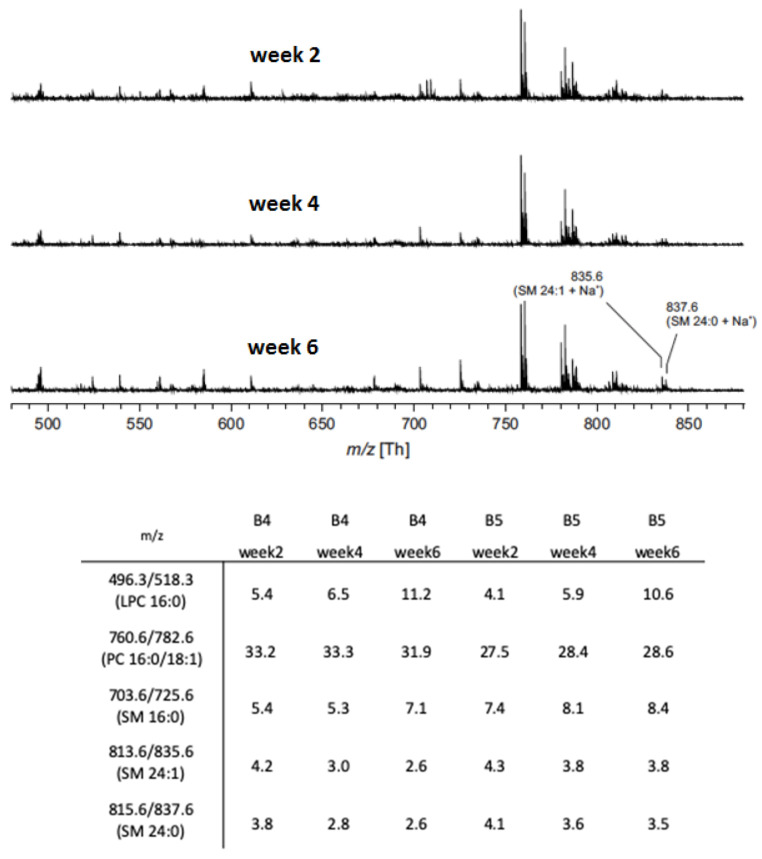
Storage-associated increase in lyso-phosphatidylcholine during microvesicle generation. All data were obtained from the positive ion MALDI-TOF mass spectra of microvesicles isolated from two RBC concentrates (B4 and B5) after two, four, and six weeks of storage (upper panel). The intensities of all detected peaks (either phosphatidylcholine (PC) or sphingomyelin (SM)) were added and the sum of the proton and sodium adducts of selected lipids were divided by this value (lower panel). Only peaks with at least 0.1% of the intensity of the base peak were used.

**Figure 4 proteomes-08-00006-f004:**
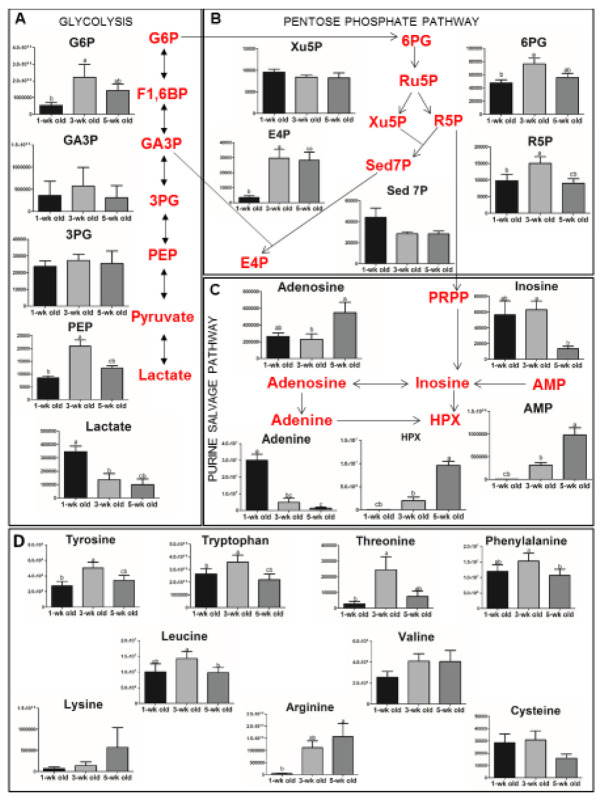
Metabolite alterations in microvesicles during storage. (**A**) Time-course changes in metabolites of the glycolysis pathway; (**B**) Time-course changes in metabolites of the pentose phosphate pathway; (**C**) Time-course changes in metabolites of the purine salvage pathway; (**D**) Time-course changes in amino acids.

**Figure 5 proteomes-08-00006-f005:**
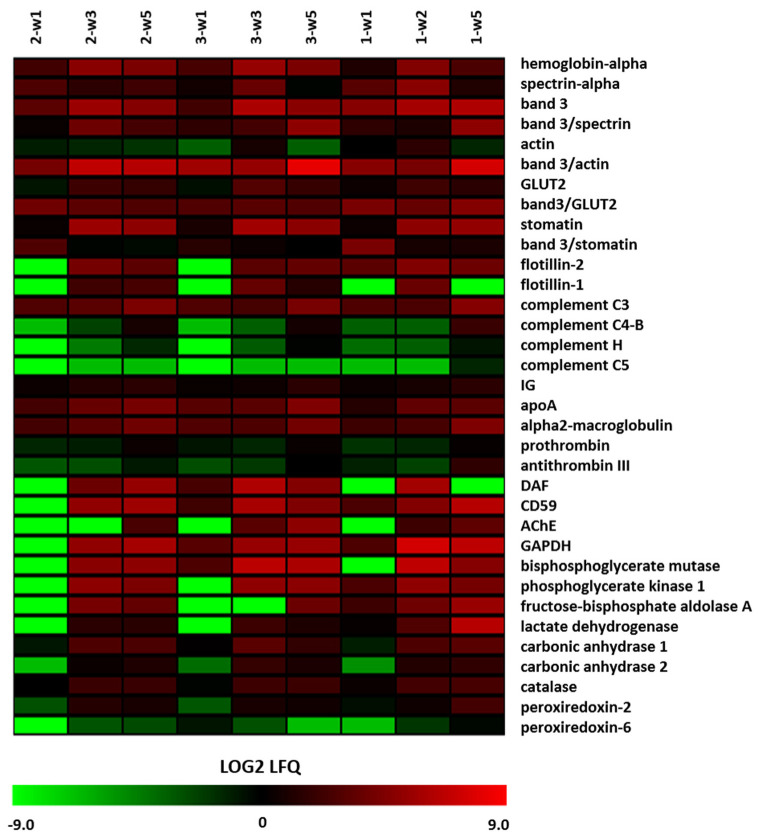
Alterations in the microvesicle proteome during storage. The proteins in the microvesicles obtained after one, three, and five weeks of storage were identified and quantified as described in the Materials and Methods. The figure shows the selected proteins for various localizations/functions (hemoglobin and integral membrane proteins, cytoskeletal/membrane-associated proteins, GPI-linked proteins, plasma proteins involved in removal, metabolic enzymes, proteins involved in coagulation, proteins involved in redox status). LFQ, label-free quantification. The different colors show the values of three biological replicates, which partially overlap with the samples used for the other analyses (Figure 1, Figure 2, Figure 3 and Figure 4). Note the different scales of the ordinates.

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
