# Peer review of "Vesiculation of Red Blood Cells in the Blood Bank: A Multi-Omics Approach towards Identification of Causes and Consequences"

_proteomes, 2020, doi:10.3390/proteomes8020006_

Round 1

Reviewer 1 Report

Summary:

The authors conducted a useful study to understand the changes in microvesicle distribution and composition of adult mature Red Blood Cells in response to cold storage. The study provides useful information about the storage lesion, which could lead to better insights into RBC storage and transfusion.

In general, it is an interesting study that generates insights from diverse points of view (image, proteomics etc), but most of the studies are relatively small in sample size. This study provides a leaping off point for larger more targeted studies.

A few comments on the paper.

The authors say that the observed vesicles are all MVs, which to me implies ectosomes, but the range of sizes vesicles include the normal size range of exosome. Are all the vesicles seen in this study ectosomes or are any exosomes? Can you tell which are exosomes and which ectosomes? Could a surface membrane protein like Transferin (https://www.ncbi.nlm.nih.gov/pubmed/30232403), Band 3 or cd59 differentiate ectosomes and exosomes? Some discussion of exosome vs ectosome would be helpful

The author note a change in distribution of the size of the MVs, it would be good to have a visual depiction (density plot or histogram or similar) of the sizes of the MVs seen. Possibly including information, if available, on if the MVs are PS positive to see if size of MV is associated with the presence of PS and possible Band 3 and CD59.

Minor points:

For the 5 standard RBC concentrates are the clinical characteristics of the donors available and if, so do these factors (age, sex/gender, ferritin level, and donation history) any of these factors affect the results?

In figure 4, parts B and C are very small and hard to read. Also in Figure 4, some of the metabolites like tryptophan and L-Tyrosine have single letters a and c without a paired letter to show which groups are statistically different.

In general, it is not clear what degree of adjustment was made for multiple testing, per legend figure 4.

In figure 2, were the statistical analyses conducted on the percentages or a logit or arsin transform of the percentages? Or if you have the actual counts underlying the percentages, a logistic regression model could be use.

Sample labels on figure 3 and 5 are different form.

I think the samples used for the data in figure 3 and 5 are non-overlapping based on labels, but that is unclear. Please clarify.

In Figure 5 the row legends have some extraneous “(“ and “)”. Capitalization in figure is also inconsistent.

Author Response

A few comments on the paper.

The authors say that the observed vesicles are all MVs, which to me implies ectosomes, but the range of sizes vesicles include the normal size range of exosome. Are all the vesicles seen in this study ectosomes or are any exosomes?

Can you tell which are exosomes and which ectosomes? Could a surface membrane protein like Transferin (https://www.ncbi.nlm.nih.gov/pubmed/30232403), Band 3 or cd59 differentiate ectosomes and exosomes? Some discussion of exosome vs ectosome would be helpful

The terminology is not unequivocal. It is likely that, in the first few days in the blood bank, the (very few) reticulocytes that were present at the time of collection, mature with the production of exosomes. Later, the RBCs will form what is usually called microvesicles, which are actually ectosomes. We have added these considerations to the Discussion (first paragraph, page 13).

The author note a change in distribution of the size of the MVs, it would be good to have a visual depiction (density plot or histogram or similar) of the sizes of the MVs seen. Possibly including information, if available, on if the MVs are PS positive to see if size of MV is associated with the presence of PS and possible Band 3 and CD59.

We have made several graphic representations of these data, but concluded that these had no additional information value. The (considerable) heterogeneity also precluded a quantitative correlation analysis of MV size with PS, band 3 and CD59 exposure on our samples. We have added this to the Discussion (second paragraph, page 13).

Minor points:

For the 5 standard RBC concentrates are the clinical characteristics of the donors available and if, so do these factors (age, sex/gender, ferritin level, and donation history) any of these factors affect the results?

This is an interesting point that we and others have addressed previously. The donors were regular blood bank donors; this has been added to the Materials and Methods section. The literature data provide no clues to any relationship between age, gender or other personal characteristics and the regular blood bag parameters. The number of samples of this study was not large enough to conclude anything on a correlation with clinical/donor characteristics and the omics data.

In figure 4, parts B and C are very small and hard to read. Also in Figure 4, some of the metabolites like tryptophan and L-Tyrosine have single letters a and c without a paired letter to show which groups are statistically different.

In general, it is not clear what degree of adjustment was made for multiple testing, per legend figure 4.

Following the reviewer’s suggestion, the figure layout has been improved by increasing the font size of labels. Moreover, a better explanation about the presentation of statistics results has been provided in the figure legend. Also, the degree of adjustment has been added.

In figure 2, were the statistical analyses conducted on the percentages or a logit or arsin transform of the percentages? Or if you have the actual counts underlying the percentages, a logistic regression model could be use.

The statistical analyses were conducted on the percentages. This has been added to the legend.

Sample labels on figure 3 and 5 are different form.

I think the samples used for the data in figure 3 and 5 are non-overlapping based on labels, but that is unclear. Please clarify.

The sample labels of figure 3 and 5 have different formats, as they were non-overlapping. This has been clarified in the legend to Figure 5.

In Figure 5 the row legends have some extraneous “(“ and “)”. Capitalization in figure is also inconsistent.

These inconsistencies have been corrected.

Reviewer 2 Report

This paper describes an intensive study of the microvesicles in the supernatant of 5 units of human red blood cell concentrates stored in CPD/SAGM and sampled 4 times over the course of storage.  They use several versions of electron microscopy, flow cytometry, proteomics, metabolomics, and lipidomic to characterized the microvesicles. The paper describes the techniques well and uses them appropriately. The conclusions are appropriate to the data. 

Major concerns:

  1. Microvesicles are actively made in response to cellular events, a) the transition from reticulocytes to erythrocytes with the loss of 50 fL of volume, b) in response to pH and electrolyte changes, c) in response to lipid and protein damage, and d) in programmed cell death. There may be multiple mechanisms forming microvesicles as mentioned in the introduction.  Measuring only three time points creates only two vectors of change and probably misses some of what is going on.
  2. The vesicles are made by stored cells, but we have no measures of those stored cells such as their pH, ATP concentrations, and the like, which are known to be important and quite variable.

Since the major conclusion of the study is that oxidative damage to proteins is the major driver of vesiculation and it is strongly associated with pH through the redox potential of glutathione.  The paper needs to acknowledge these weaknesses

Minor concerns:

  1. Hemoglobin and haptoglobin mentioned on page 11, line 261, do not appear in Figure 5 as stated.

Author Response

Major concerns:

  1. Microvesicles are actively made in response to cellular events, a) the transition from reticulocytes to erythrocytes with the loss of 50 fL of volume, b) in response to pH and electrolyte changes, c) in response to lipid and protein damage, and d) in programmed cell death. There may be multiple mechanisms forming microvesicles as mentioned in the introduction.  Measuring only three time points creates only two vectors of change and probably misses some of what is going on.

Indeed, it is likely that multiple mechanisms are involved in microvesicle formation, as discussed throughout the Discussion section. We have now added the maturation of reticulocytes (see also our answers to reviewer 1), and a short justification on our selection of the time points, especially the week 3 time point (Materials and Methods, first paragraph, page 3 and Discussion, first paragraph, page 13).

  1. The vesicles are made by stored cells, but we have no measures of those stored cells such as their pH, ATP concentrations, and the like, which are known to be important and quite variable.

Since the major conclusion of the study is that oxidative damage to proteins is the major driver of vesiculation and it is strongly associated with pH through the redox potential of glutathione.  The paper needs to acknowledge these weaknesses

The storage-related values of pH, ATP, etc. may be variable, but the blood bank literature shows that the changes in these values are very consistent  between the storage periods we have analyzed. We have added this consideration to the justification of our time points, based on previous (proteomics and other) literature data (also see our answer to the comment above). The relatively low number of donors/RBC concentrates we could analyze in this multi-omics study was too low for a conclusive correlation analysis of our data with the standard blood bank parameters (see also our reply to the first minor point of Reviewer 1. We have added this consideration to the conclusion part of the Discussion (page 16).  

Minor concerns:

  1. Hemoglobin and haptoglobin mentioned on page 11, line 261, do not appear in Figure 5 as stated.

Hemoglobin appears in Figure 5 as Hb; this has been changed to ‘hemoglobin’; haptoglobin can be found in the complete proteome list of the supplemental file. This has been corrected in the text.